# The Role of SIRT1 on DNA Damage Response and Epigenetic Alterations in Cancer

**DOI:** 10.3390/ijms20133153

**Published:** 2019-06-28

**Authors:** Débora Kristina Alves-Fernandes, Miriam Galvonas Jasiulionis

**Affiliations:** Department of Pharmacology, Escola Paulista de Medicina, Universidade Federal de São Paulo, São Paulo 04039-032, Brazil

**Keywords:** SIRT1, DNA damage/repair, epigenetics, cancer development

## Abstract

Sirtuin-1 (SIRT1) is a class-III histone deacetylase (HDAC), an NAD+-dependent enzyme deeply involved in gene regulation, genome stability maintenance, apoptosis, autophagy, senescence, proliferation, aging, and tumorigenesis. It also has a key role in the epigenetic regulation of tissue homeostasis and many diseases by deacetylating both histone and non-histone targets. Different studies have shown ambiguous implications of SIRT1 as both a tumor suppressor and tumor promoter. However, this contradictory role seems to be determined by the cell type and SIRT1 localization. SIRT1 upregulation has already been demonstrated in some cancer cells, such as acute myeloid leukemia (AML) and primary colon, prostate, melanoma, and non-melanoma skin cancers, while SIRT1 downregulation was described in breast cancer and hepatic cell carcinomas. Even though new functions of SIRT1 have been characterized, the underlying mechanisms that define its precise role on DNA damage and repair and their contribution to cancer development remains underexplored. Here, we discuss the recent findings on the interplay among SIRT1, oxidative stress, and DNA repair machinery and its impact on normal and cancer cells.

## 1. Introduction

### 1.1. DNA Damage Signaling Response

Oxidative stress is a result of an imbalance between the generation of reactive oxygen species (ROS) and the antioxidant activity of cells [1,2]. The basal levels of ROS are involved with the activation of cell proliferation, survival, differentiation, motility, and stress-responsive pathways, whereas increased levels of ROS indicates oxidative damage to DNA, proteins, and lipids. However, if the oxidative DNA damage is not properly repaired, it may cause mutations and promote carcinogenesis [1,2,3,4,5]. When the excessive production of ROS leads to extensive and irreparable DNA damage, such as base modifications, DNA cross-links, single-strand (SBs), or double-strand breaks (DSBs), it promotes cell death [6,7]. Furthermore, other forms of DNA damage, such as DNA adducts and abasic sites, may also be induced by a large diversity of environmental stimuli and intrinsic factors [8]. The 8-hydroxy-2′-deoxyguanosine (8-OHdG) is a DNA-adduct commonly associated with oxidative stress [1]. The ability of ROS to damage cells and mutate DNA correlates with cellular aging and it has been associated with the initiation, promotion, and progression of cancer, which demonstrates the importance of these mechanisms in the context of cancer development [1,9]. Cells have mechanisms to limit genomic instability via DNA the damage response (DDR), which may remove or tolerate these DNA lesions [10]. Depending on the extent of the DNA damage, diverse intracellular responses might be induced, such as cell cycle arrest, activation of DNA repair machinery, and cell death [11]. 

DNA adducts demonstrate toxicity related to the polymerase-stalling events during DNA replication or transcription [12]. During DNA synthesis, the presence of small DNA adducts, such as O^6^-alkylguanine and N7-alkylguanine, may interfere in the DNA polymerase activity if the DNA repair pathways (mismatch repair (MMR) and base excision repair (BER)) are working. Moreover, when the replication fork is blocked, the collapse in the fork may create a toxic one-ended DNA DSB and consequently, the events in response to DNA damage may be initiated [13,14,15]. A DSB is a very potent lesion which occurs when both complementary DNA strands break within a few base pairs of distance, leading to the dissociation of both DNA ends, resulting in DNA repair or in recombination errors [16]. Homologous recombination (HR) and non-homologous end joining (NHEJ) are two different pathways to repair DSBs and different conditions lead to distinct pathways. HR-mediated repair needs a homologous sequence on the sister chromatid as a template to correctly restore the damaged DNA site [17] and NHEJ repair ligates DSBs directly, mediating an error-prone repair process [18,19]. 

Other types of DNA damage which can be induced by UV-light radiation, include DNA photoproducts, such as *cis-syn* cyclobutane pyrimidine dimers (CPD), pyrimidine (6-4) pyrimidine photoproducts (6-4PP), and cisplatin-induced DNA intra-strand crosslinks *cis*-Pt(NH3)2d(GpG) (Pt-GG) [20]. Nucleotide excision repair (NER) is the major mechanism involved in the removal of a wide range of bulky DNA damage, such as UV-photoproducts, chemical adducts, and some types of DNA crosslinks. Global genome NER (GG-NER) and transcription-coupled NER (TC-NER) are NER subpathways with at least 30 proteins involved in recognition, dual incision, repair synthesis, and ligation steps [21].

It is important to remember that NHEJ, HR, MMR, BER, NER, and protein-linked DNA break (PDB) repair in combination with DDR signaling and damage tolerance pathways are mechanisms to cell survival, however, their interaction can switch off repair and trigger cell death [12]. The major sensors from DDR are the MRE11-RAD50-NBS1 (MRN) complex, which detects DSBs; and replication protein A (RPA) and RAD9-RAD1-HUS1 (9-1-1) complex, which detects DNA SBs. The MRN complex recruits ataxia-telangiectasia mutated (ATM), while RPA and the 9-1-1 complex mobilize ataxia-telangiectasia and Rad3-related (ATR), which is bound by their interaction with ATR-interacting protein (ATRIP). Then, these proteins phosphorylate the histone variant H2AX on Ser139 (known as γH2AX) in the region close to the DNA damage, required as a mediator for DNA damage checkpoint 1 (MDC1) protein to sustain and amplify DDR signaling by increasing the accumulation of MRN complex and ATM activation. ATM and ATR phosphorylate BRCA1, which is recruited to the DNA damage site, and p53-binding protein 1 (53BP1) are also involved in the maintenance of DDR signaling. Sometimes, the diffusible kinases, CHK2 (cell cycle checkpoint kinase 2—phosphorylated by ATM) and CHK1 (phosphorylated by ATR), may signal for downstream effectors, such as p53 and CDC25 phosphatase (cell division cycle 25), to act away from the damaged DNA, leading to either cell cycle arrest followed by DNA damage repair, senescence generated by unrepaired DNA damage, or cell death by apoptosis [22,23,24,25]. 

### 1.2. Epigenetic Effects of DNA Damage

Besides the repair mechanisms, another important issue impacted by DNA damage is the chromatin state, which may affect the sensitivity of DNA to DNA-genotoxic agents, as well as interfere in the access for DDR signaling and DNA repair factors [17]. The major chromatin alteration related to DNA repair is γH2AX [26]. Histone acetylases (HATs) and deacetylases (HDACs) can also identify DNA damage sites, contributing to the access of DNA repair proteins, silencing transcription during the repair process, restoring the chromatin state after repair, and shutting off DDR [27,28]. 

HDACs are major agents of epigenetic regulation and their dysfunctional deacetylase activity has been strictly related to the tumorigenesis process and cancer metastasis [29]. In this regard, mammalian sirtuins (SIRT 1 to 7) belonging to NAD+-dependent class III HDACs, have different modes of action, targets, and subcellular compartments. SIRT1, SIRT6, and SIRT7 are found in the nucleus, thus collaborating with epigenetic regulation of the cell as these enzymes target mainly histones and transcription factors. SIRT3, SIRT4, and SIRT5 are mitochondrial proteins, and SIRT3 specifically, has been reported to protect the mitochondrial DNA and to prevent apoptosis by epigenetic mechanisms [30,31,32,33]. SIRT2 can be found in the nucleus or cytoplasm, and this shuttle depends on the phase of the cell cycle [30,31]. Sirtuins are involved in the prevention of DNA damage and promotion of DNA repair by many different pathways. As an example, DNA damage can be prevented by the suppression of ROS in the mitochondria or by the activation of ROS-neutralizing enzymes [31,34,35,36,37,38]. 

The most studied sirtuin is SIRT1, which has both histone and non-histone proteins as targets. SIRT1 is widely recognized as a crucial epigenetic regulator involved in many biological processes, including metabolism, genomic stability maintenance, reprograming, aging, and tumorigenesis, among others [39,40,41]. SIRT1 expression and its deacetylase activity are controlled in normal cells by other proteins, such as p53, testis-specific protein Y-encoded-like 2 (TSPYL2), CHK2, hipermethylated in cancer 1 (HIC1), and cell cycle apoptosis regulator 2 (CCAR2 or DBC1) [40,41,42,43,44]. 

SIRT1 modulates the chromatin state, interfering mainly in the transcription process. The expression or silencing of a distinct set of genes includes many posttranslational modifications of histone and non-histone proteins and the harmony among these modifications controls the transcription process. Reversible histone modifications include acetylation, methylation, phosphorylation, ubiquitination, ADP-ribosylation, carbonylation, butyrylation, and propionylation [45,46], and the enzymes regulating these processes are very important to the maintenance of the structure and organization of the chromatin, allowing a dynamic epigenetic gene regulation [46,47]. Histone phosphorylation plays a role in DNA damage repair and has been associated with cell cycle modifications, while histone methylation and acetylation are essential for secondary structure formation and epigenetic silencing [46]. It has already been demonstrated that the downregulation of DNA repair genes in response to oxidative stress is independent of DNA methylation, evidencing the role of histone acetylation as an epigenetic mechanism to suppress specific genes in response to DNA damage [48]. 

SIRT1 is responsible for the histone deacetylation at H1 lysine 26 (H1K26ac), H3 lysine 9 (H3K9ac), and H4 lysine 16 (H4K16ac) [49]. Recently, H4 lysine 4 (H4K4ac) was identified as a new target of SIRT1 in breast cancer [50]. Acetyl H4K16 is an important residue for chromatin formation and its deacetylation is associated with tight chromatin compaction. SIRT1 knockdown showed a considerable effect on persistent H4K16ac after DNA damage [51]. Besides that, SIRT1 induces the production of H3 trimethyl K9 (H3K9me3) and H4 monomethyl K20 (H4K20me3), two histone marks associated with repressed chromatin. SIRT1 also induces the production of H3 dimethyl K79 (H3K79me2), a barrier that separates active and inactive chromatin domains [49]. It was shown that SIRT1 recruits and, by its deacetylase activity, directly activates the suppressor of variegation 3-9 homologue 1 (SUV39H1), a histone methyltransferase responsible for the accumulation of histone H3 containing a trimethyl group at its lysine 9 position [49]. These data are supported by SIRT1^−/−^ mice, which showed an altered pattern of histone modification with decreased K9me3 and increased acetylation of H3K9, which is related with embryonic lethality [52]. 

The role of SIRT1 in cancer seems contradictory. SIRT1 overexpression has been described in many cancers compared to normal tissues, including leukemia [53], non-melanoma [51] and melanoma skin cancer [54,55], prostate [56], and colon carcinomas [57]. On the other hand, SIRT1 down regulation has also been observed in other neoplasms, such as breast cancer and hepatic cell carcinomas [52]. Since SIRT1 can play different roles depending on its subcellular localization (in both nucleus and cytoplasm) and tissue [58], it becomes reasonable to find its diverse effect on different types of tumors. 

Although SIRT1 participates in diverse cellular processes, the mechanisms underlying its precise role on DNA damage and repair and its contribution to cancer development remains underexplored. Here, we present evidences and discuss the recent findings on the interplay among SIRT1, oxidative stress, and DNA repair machinery and its impact on normal and cancer cells.

## 2. SIRT1 as an Epigenetic Regulator in Response to DNA Damage

SIRT1 has been described as an important player in the DNA damage response, both as a histone deacetylase at DNA damage sites and as a deacetylase of proteins involved in DNA repair and DDR [28,59,60,61,62]. The histone deacetylation results in alterations of chromatin compaction, while non-histone protein deacetylation is related to protein activity [61]. Both modifications catalyzed by SIRT1 have a role in the protection against DNA damage as well as in the response to this damage. 

The cell sensitivity to the DNA damage and the repair mechanisms can be modified by chromatin compaction, working as a barrier to many DNA templated processes, including DNA repair [63]. It has been described that SIRT1 impairs the formation of UV-induced 6-4PP and Pt-GG photoproducts in human fibroblasts by the maintenance of a highly condensed heterochromatin. Thus, CPD, which generates a minor distortion in the double helix of DNA, was detected in both euchromatin and heterochromatin while 6-4PP and Pt-GG, major modifications of the double helix structure, could be found only in euchromatin, demonstrating the importance of condensed chromatin in the formation of these lesions [20]. In addition, SIRT1 can contribute to decreased levels of 8-OHdG, a critical oxidative stress biomarker in the hippocampus of rats treated with a ketogenic diet [64]. This treatment leads to increased levels of NAD+, a substrate for sirtuins and poly(ADP-ribose) polymerases (PARPs), affecting different cellular functions from gene expression to DNA repair [65]. 

In response to environmental stimuli, histone modification performs one of the major epigenetic regulatory mechanisms [61], as histone and chromatin modifications influence both temporal and spatial control of DNA damage repair [66]. Genomic instability and changes in the transcriptional profile are known hallmarks of cancer and eukaryotic aging [67]. It was described that yeast Sir proteins, sirtuin orthologs, may build an inactive heterochromatin state by polymerizing across nucleosomes and thus repressing transcription [67,68,69]. In this regard, Sir proteins may shift their localization from transcriptionally repressed genes to DNA damage sites via MEC1 (ATM homolog)-mediated signaling, resulting in de-repression of genes that were epigenetically silenced by the Sir2/3/4 complex [69]. In a similar manner, another group showed that SIRT1 is required for the efficient repair of DSBs and for genome preservation in response to genotoxic stress in mammalian stem cells, where SIRT1 can dissociate from transcriptionally repressed repetitive DNA loci and relocate to DNA breaks, to promote repair via the ATM signaling pathway. This DNA damage-induced relocalization of SIRT1 impacts in chromatin reorganization, resulting in transcriptional changes associated with age-related modifications [67]. 

When SIRT1 is located on the gene promoter, it may induce silencing by the recruitment of DNMTs to the damaged site, increasing methylation on this promoter region [22,70]. In this regard, SIRT1 knockdown or SIRT1 downregulated by a combined treatment (Resveratrol + Pterostilbene) produced decreased γH2AX and reduced DNMT1, DNMT3a, and DNMT3b in breast cancer cells [71]. After oxidative DNA damage induced by treating colorectal and colon cancer cell lines with H_2_O_2_, the heterodimer, MSH2-MSH6, binds with a high affinity to chromatin and then recruits DNA methyltransferase 1 (DNMT1) and PCNA at the DNA damaged site [62]. Previous work from this group also demonstrated that key epigenetic proteins—SIRT1, enhancer of zeste homologue 2 (EZH2), DNMT1, and DNMT3B—are recruited to DSB, and the emergence of repressive histone modifications, such as hypoacetyl H4K16, H3K9me2/me3, and H3K27me3, occurs to establish and maintain chromatin compaction around the damage site. Moreover, the authors showed that, although this chromatin repression is transient during DNA repair, a small percentage (0.9%) of cells retain a stable silencing of an associated promoter, suggesting that sustained DNA damage repair processes could be a risk for tumor development [60]. Mechanistically, DNMT1 becomes part of a silencing complex with DNMT3B and polycomb repressive complex 4 (PRC4 components—SIRT1 and EZH2) that co-localizes with γH2AX to form DNA damage-induced foci. While histone marks and nascent transcription changes were observed in high expression genes enriched for PRC4 complex members, the gain of promoter DNA methylation on CpG island-containing low expression genes suggests transcriptional silencing and cancer-specific DNA methylation alteration [28]. Ding and coworkers [62] still suggest that DNMT1 works as a scaffold to bring SIRT1, EZH2, and DNMT3B to DNA with oxidative damage to induce the transcriptional repression of genes with CpG island-containing promoters not as a result of DNA methylation, but through a chromatin-based mechanism. They showed that catalytically inactive DNMT1 was recruited to chromatin and cooperated with MSH2/6 proteins in a way very similar to wild type (WT) DNMT1 cells [62]. In another study, our group has demonstrated the role of SIRT1 with DNMT3B co-participation in *Mxd1* epigenetic silencing in a mouse melanoma model based on cellular stress [55]. MXD1 is a MYC inhibitor and competes with MAX, a co-activator, with this MYC/MXD1/MAX signaling pathway being implicated in the development of some types of cancer [72,73]. 

SIRT1 was also shown to act as an important remodeler of chromatin affinity in human colorectal cancer cells (HCT116) exposed to oxidative stress. In this model, besides deacetylating H4K16ac, SIRT1 was able to deacetylase hMOF (human MOF), a member of the HAT family, which affects hMOF recruitment and leads to the downregulation of important genes required for DNA double strand breaks repair, such as BRCA2, RAD50, and FANCA [48]. Another important gene regulated in response to continuous DNA breaks is *arf*, which has been shown to be transcriptionally controlled by SIRT1 along with transcription factor E2F1, both involved in the signaling cascade initiated by PARP1, a molecular sensor of SBs [74]. ARF is a tumor suppressor involved in the p53 regulation and *arf* gene is frequently inactivated in diverse human cancers [75,76]. E2F1 also is a target of SIRT1 by deacetylation and it is associated with a failure to stimulate important genes to induce apoptotic pathways [77].

SIRT1 has also been implicated in the recruitment of important DNA repair proteins, such as Nijmegen breakage syndrome 1 (NBS1) and Rad51 in keratinocytes containing episomes of high-risk human papillomaviruses (HPV) types contributing to viral activities, such as genome maintenance, amplification, and gene transcription [78]. 

## 3. Other Forms by which SIRT1 Modulates the DNA Repair Response

SIRT1 has a key role regulating the cellular fate under stress through its involvement with p53-dependent aging, cancer, and cellular reprogramming [41]. Importantly, if cells allow the repair machinery either to act by cell cycle arrest or if they trigger programmed cell death as a result of extensive and unrepairable damage, it will be decided by the balance between p53 and DNA repair proteins (reviewed in [41]). The increase of SIRT1 deacetylase activity in response to DNA damage is related to an increase in the risk of cancer in mammals, due to the effect on the inhibition of p53 and other tumor suppressor genes, such as retinoblastoma (Rb) [79].

SIRT1 is directly related to the DNA damage response and can recruit repair proteins to DSBs’ sites and modulate their activity by deacetylating them. SIRT1-mutated MEFs (mouse embryonic fibroblast) were shown to be more sensitive to UV radiation and γ-irradiation, as SIRT1 can modulate the γH2AX, Rad51, BRCA1, and NBS1 foci formation upon DNA damage, interfering in DNA repair and the cell cycle checkpoint [52]. The key role of SIRT1 in regulating the activity of the DNA repair complex at DSB site has been reported by many authors. Repair proteins, such as KU70 and the FOXO family, were shown to be deacetylated and activated by SIRT1 upon DNA damage [59,80,81,82,83,84]. KU70 is a crucial protein belonging to NHEJ, the major DNA repair mechanism for DSB in mammalian cells [85]. NHEJ repair initiates after the binding of the KU70/KU80 heterodimer to broken DNA strands. The absence of these factors prejudice DSB repair, increasing cell sensitivity for radiation [86,87]. Besides that, the interaction between SIRT1 and KU70 is site-competitive with LSD1 (lysine-specific demethylase 1). The increased SIRT1–KU70 interaction can favor DNA repair by opening the chromatin, and this interaction can increase the likelihood of BCR-ABL mutations in chronic myeloid leukemia (CML) [88,89], and has been involved in resistance to treatment with tyrosine kinase inhibitors [90,91]. On the other hand, when the LSD1–KU70 interaction is favored, there is less chromatin opening and the repair is then decreased, resulting in increased DNA damage and cell death [72]. More recently, Yu and collaborators [92] showed that KU70 also controls SIRT1 activity through SHP-1, a dephosphorylating enzyme member of the protein tyrosine phosphatase (PTP) family. Thus, KU70 inhibition was shown to reduce DNA repair efficiency by both NHEJ and HR pathways in adult T-cell leukemia-lymphoma (ATL) cells and the tumorigenesis ability was impaired in Jurkat-xenografted mice after KU70 silencing [92]. 

In nitric oxide-induced stress, pancreatic β-cells showed more DNA damage, such as comet tail formation, when in the absence of functional SIRT1. This was attributed to SIRT1-dependent FoxO1 (Forkhead transcription factor) activity [81]. FoxO1 is maintained in the cytoplasm by AKT phosphorylation activity according to the growth factors supply [82]; however, in a stressful situation, FoxO1 may be translocated to the nucleus, stimulating the transcription of repair genes, such as GADD45α [81]. SIRT1 can activate the FOXO family by deacetylation, favoring the expression of protective target genes involved in DNA repair [82,83,84], while the deacetylation of both p53 and FOXO proteins reduces apoptosis [84]. SIRT1 and the FOXO family (FOXO3 and FOXO4) form a complex that induces cell cycle arrest and resistance to oxidative stress, but also attenuates the FOXO-induced apoptosis [82,83]. FOXO proteins have been identified as tumor suppressors and are linked to chromosomal translocations in many cancers [93,94,95,96].

SIRT1 and ATM have a synergistic relationship, where SIRT1 is recruited to DNA breaks in an ATM-dependent manner, while SIRT1 also deacetylates ATM, and it stimulates its activity by auto-phosphorylation and stabilizes the ATM at DSB sites [97]. In the case of DNA damage, ATM and ATR cooperate with transducer kinases and mediator proteins, phosphorylating many substrates causing a proper response to lesions, such as DNA repair, senescence, or cell death [98]. SIRT1 binds and promotes NBS1 deacetylation, which can then be phosphorylated at Ser343 by ATM in response to IR, promoting the DNA repair signaling pathway and cell cycle checkpoint. NBS1 associates with MRE11 and RAD50 to form the MRN complex involved in the detection and response to DNA damage [81,99]. The deacetylation of Werner syndrome protein (WRN) by SIRT1 was described as facilitating the response since it can interact with diverse DNA repair proteins, including Ku complex, MRN complex, PARP-1, RAD52, and p53 [100,101,102]. Kruppel-associated box (KRAB)-associated protein 1 (KAP1) is also phosphorylated (Ser824) by ATM, but the deacetylation by SIRT1 stabilizes its interaction with 53BP1, another well-known marker of DSBs, inducing NHEJ repair and inhibiting HR repair pathways. KAP1 is a regulator of genomic stability, heterochromatin formation, and target gene silencing [103]. In addition, SIRT1 favors NER repair by deacetylating XPA (xeroderma pigmentosum group A) protein after UV radiation. NER is the main repair process for UV-induced DNA damage [21]. The deacetylation of XPA at residues K63, K67, and K215 is crucial to promote interaction and phosphorylation by ATR, therefore increasing cAMP-enhanced NER [104]. 

SIRT1 was also described as a critical regulator of multiple DNA repair enzymes belonging to MMR and BER systems after oxidative stress, such as MHS2, MSH6, and apurinic/apyrimidinic endonuclease APEX1, thus maintaining the genomic fidelity and integrity in human embryonic stem cells. Although the mechanisms by which SIRT1 controls these enzymes are not fully understood, the p53 activity does not seem to be related to the initial events (2 h), when a decrease in DNA repair proteins and induction of DNA damage (DSBs) are observed, but it is involved with later events, such as the induction of DSB-mediated apoptosis [11]. APEX1 plays an essential role in the repair of single-strand DNA breaks and abasic DNA sites generated spontaneously or chemically [105], and after genotoxic stress, it is targeted by SIRT1’s deacetylase activity at lysine 6 and 7, stimulating the binding of APEX1 to BER protein X-ray cross-complementing-1 (XRCC1) in the DNA damage site [106]. XRCC1 serves as a scaffold protein to APEX1 and other proteins of the BER complex and stimulates the AP endonuclease activity helping on DNA repair [107]. 

## 4. Concluding Remarks

Although SIRT1 has been implicated in diverse cellular processes affecting senescence, aging, proliferation, inflammation, energy homeostasis, gene silencing, and extended lifespan, among others, the interplay between SIRT1 and DNA repair in normal and cancer cells is not completely understood and should be better explored [40,41,108,109,110]. 

Hence, this review presented evidence for the essential role of SIRT1 in the reduction of DNA damage (photoproducts and oxidative DNA damage) by contributing to heterochromatin formation, and by recruiting and activating many DNA repair factors to the DNA damaged site. Inevitably, SIRT1 seems to be a major epigenetic regulator of the transcriptional profile and genomic stability. Upon stress, SIRT1 can be relocate to DNA damaged sites, favoring abnormal epigenetic marks and culminating in the expression of transcriptionally repressed DNA loci, such as aging- and tumor-related genes, which might favor the development and progression of tumors. Besides that, the interaction with HATs and DNMTs also justifies its position as a master epigenetic regulator (Table 1 and Figure 1). 

SIRT1 interacts with distinct proteins from the main DNA repair mechanisms (HR, NHEJ, MMR, NER, BER) and DDR pathways recruiting them to DNA damage foci or activating the proteins involved in DNA repair by deacetylating them. These processes aim to help the cells to live without damaged DNA, but are also subject to error, leading to mutations and abnormal epigenetic marks. Thus, SIRT1’s role as a DNA repair promoter and guardian of genomic stability is clear, but its relationship with tumorigenesis induction is not fully understood and studies that elucidate these pathways will provide a breakthrough in cancer biology. 

## Figures and Tables

**Figure 1 ijms-20-03153-f001:**
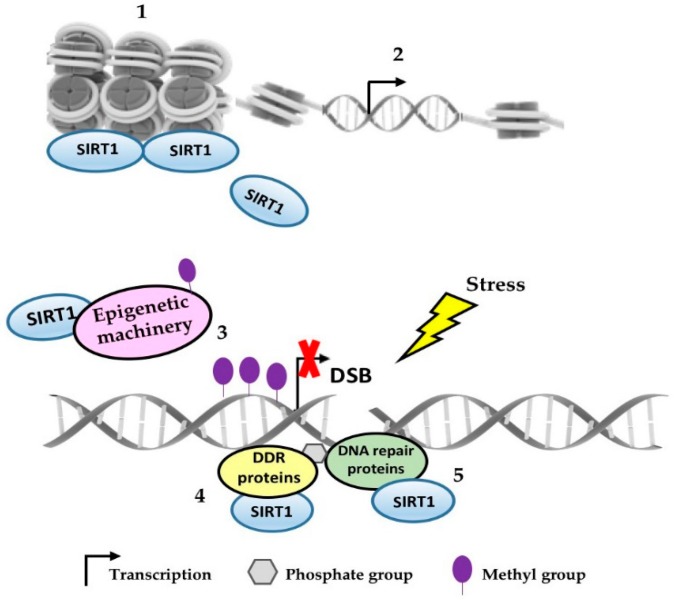
SIRT1 as an epigenetic regulator and DNA repair response modulator. SIRT1 contributes to heterochromatin formation and can be associated with transcriptionally repressed repetitive DNA loci (**1**); upon stress, SIRT1 can be dissociated from these regions towards the DNA damage site, resulting in chromatin reorganization which favors transcriptional changes (**2**); SIRT1 recruits epigenetic machinery to the transcriptional silencing around the DNA damage site (**3**); SIRT1 also participates in the recruitment and deacetylation of DDR (**4**) and DNA repair proteins, (**5**) helping in many steps of repair pathways.

**Table 1 ijms-20-03153-t001:** The role of SIRT1 and its effects in response to DNA damage.

Role of SIRT1	Effects	Cell or Tissue Type	Reference
**Protection against DNA damage**	Absence of 6-4PP and Pt-GG in heterochromatin associated with SIRT1	Human fibroblasts	[20]
	Decreased levels of 8-OHdG after the increase in SIRT1 activity and mRNA level	Rat hippocampus	[64]
**Regulation of genomic stability and transcriptional changes**	Relocalization of Sir2/3/4 and de-repression of epigenetically silencing genes in response to DNA damage	Yeast	[69]
	Relocalization of SIRT1 to DSBs followed by transcriptional changes	Mammalian stem cells	[67]
	Recruitment of key epigenetic proteins (DNMTs, EZH2) to DNA damage site	Normal and cancer cells	[28,55,60,62,71]
	Deacetylation of HAT (hMOF) and E2F1 leading to downregulation of genes involved in DNA repair and genomic stability	Cancer cells	[48,74]
	Recruitment of DNA repair proteins contributing with viral activity, including gene transcription	Keratinocytes containing HPV episomes	[78]
**Modulation of DDR and DNA repair**	Deacetylation of p53 interfering in cell death	Normal and cancer cells	reviewed in [41]
	Deacetylation of γH2AX, Rad51, BRCA1 and NBS1, regulating the foci formation	MEFs	[52]
	Interaction with Ku70 protein belonging to NHEJ	Cancer cells	[88,89,90,91,92]
	Deacetylation of FOXO family favoring the expression of target repair genes, cell cycle arrest and resistance to oxidative stress	Normal cells	[81,82,83,84]
	Regulation of many important proteins to DDR and DNA repair, including ATM, NBS1, WRN, KAP1, XPA, MSH2, MSH6, APEX1	Normal and cancer cells	[11,97,99,100,101,102,103,104,106]

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
