# Peer review of "The Role of SIRT1 on DNA Damage Response and Epigenetic Alterations in Cancer"

_ijms, 2019, doi:10.3390/ijms20133153_

Round 1
Reviewer 1 Report
The review is well written and structured but in my opinion there are some concepts that could be more explained.
For example, when you introduce SIRT1 protein, you don't talk about the others sirtuins, and in a review, it is important to explain all the general subjects. Another concept that appears several times is oxidative stress, but without explanation. Please, enlarge the review.
It would be interesting that you make a paragraph talking about the effects of SIRT3 as a master epigenetic regulator in mtDNA.
The list of references could be more extensive.
Author Response
Comments from the editor and reviewer:
We are grateful for the important suggestions, which have been addressed in the new text. Please, you will find below the detailed answers for your comments.
-Reviewer 1
The review is well written and structured but in my opinion there are some concepts that could be more explained.
For example, when you introduce SIRT1 protein, you don’t talk about the others sirtuins, and in a review, it is important to explain all the general subjects. Another concept that appears several time is oxidative stress, but without explanation. Please, enlarge the review.
The text was carefully revised and rewritten. On lines 24-36 you will find the concept about oxidative stress. The introduction about other sirtuins was presented on lines 85- 94.
It would be interesting that you make a paragraph talking about the effects of SIRT3 as a master epigenetic regulator in mtDNA.
A paragraph about SIRT3 was included on lines 88-90.
The list of references could be more extensive.
We increased the list of references with these modifications. The references number 1-9 and 30-38 were added.
Reviewer 2 Report
In the review SIRT1, a master epigenetic regulator, and its role on DNA damage response in cancer, the authors set out to summarize the role that Sirtuin-1 plays in DNA damage repair, and highlight the inconsistency of SIRT1 being alternatively upregulated or downregulated in different types of cancer. The topic is of interest to the field, and suggests questions which are worth investigating. The paper is generally well researched. However, the biggest concern is editing, grammar, and language use. While the article seems scientifically sound, the quality of the writing is a barrier to fully absorbing the manuscript. Overall, it is my opinion that with some major editing, though, this paper could be appropriate for publication.
Major concerns:
As mentioned above, the biggest concern is editing, grammar, and language use. This paper needs significant editing before it is up to standards. A partial (and in no way comprehensive) list of typographical errors are included below.
Minors concerns:
Your introduction would benefit from being separated into two key subheadings: DNA damage signaling response, and epigenetic effects of DNA damage
You switch between hyphenated (H3-K9) and non-hyphenated (H3K9) notation for histone sites. The non-hyphenated form is commonly accepted, and at the very least, you should be consistent.
Line 139: You start talking about Sir proteins: were they ever properly introduced or discussed?
Line 219: Why mention the comet assay if there is no further explanation of what that means or why it’s relevant?
Partial list of typographical errors:
Multiple sections throughout where a single word (the, a, is, etc.) is missing
Line 16: Should read: “was described in tissue such as breast cancer”
Line 18: precise role on the DNA damage
Line 27: Extent of DNA damage, not extend
Line 31: may difficult impair the DNA polymerase activity
Line 66: Besides the repair mechanism, another important issue…
Line 71: restoring the chromatin state…
Line 99: Not only, SIRT1 also
Line 115: “Here we present a lot of evidences” is grammatically incorrect and not the proper tone for a research paper.
Author Response
Comments from the editor and reviewer:
We are grateful for the important suggestions, which have been addressed in the new text. Please, you will find below the detailed answers for your comments.
-Reviewer 2
In the review SIRT1, a master epigenetic regulator, and its role on DNA damage response in cancer, the authors set out to summarize the role that Sirtuin-1 plays in DNA damage repair, and highlight the inconsistency of SIRT1 being alternatively upregulated or downregulated in different types of cancer. The topic is of interest to the field, and suggests questions which are worth investigating. The paper is generally well researched. However, the biggest concern is editing, grammar, and language use. While the article seems scientifically sound, the quality of the writing is a barrier to fully absorbing the manuscript. Overall, it is my opinion that with some major editing, though, this paper could be appropriate for publication.
Major concerns:
As mentioned above, the biggest concern is editing, grammar, and language use. This paper needs significant editing before it is up to standards. A partial (and in no way comprehensive) list of typographical errors are included below.
Minors concerns:
Your introduction would benefit from being separated into two key subheadings: DNA damage signaling response, and epigenetic effects of DNA damage
The introduction was separated into these two subheadings.
You switch between hyphenated (H3-K9) and non-hyphenated (H3K9) notation for histone sites. The non-hyphenated form is commonly accepted, and at the very least, you should be consistent.
The text was revised and non-hyphenated form was maintained.
Line 139: You start talking about Sir proteins: were they ever properly introduced or discussed?
An explanation about Sir proteins and its role on heterochromatin formation was added on lines 163-165.
Line 219: Why mention the comet assay if there is no further explanation of what that means or why it’s relevant?
The comet assay is not relevant for this review, thus we improved the explanation talking about the type of DNA damage (lines 245 and 246)
Partial list of typographical errors:
Multiple sections throughout where a single word (the, a, is, etc.) is missing
All typographical errors observed were corrected.
Line 16: Should read: “was described in tissue such as breast cancer”
Corrected - line 16.
Line 18: precise role on the DNA damage
Corrected - line 18.
Line 27: Extent of DNA damage, not extend
This error was corrected (line 39).
Line 31: may difficult impair the DNA polymerase activity
Corrected - line 43 - “may interfere in the DNA polymerase activity”.
Line 66: Besides the repair mechanism, another important issue…
Corrected - line 82.
Line 71: restoring the chromatin state…
Corrected - line 87.
Line 99: Not only, SIRT1 also
Corrected - line 125.
Line 115: “Here we present a lot of evidences” is grammatically incorrect and not the proper tone for a research paper.”
Corrected - line 141.
Reviewer 3 Report
Dear Authors,
your paper entitled "SIRT1, a master epigenetic regulator, and its role on DNA damage response in cancer" is a well-written review, complete and adequate in the cited reference work and complete in the scientific revision about the role of SIRT1 as a protetictive factor for cancer and as a tumor promoter. I suggest only few changes in english style and language:
- rephrase the concept in lanes 30-32 of Introduction ("Small DNA adducts.........................");
- the same for the phrase in lane 207 of the paragraph "Other forms by which SIRT1 modulates DNA repair response" ("The absence of these factors.......".
I also suggest a less twisted title.
After these changes the review should be accepted for publication in International Journal of Molecular Sciences.
Author Response
Comments from the editor and reviewer:
We are grateful for the important suggestions, which have been addressed in the new text. Please, you will find below the detailed answers for your comments.
-Reviewer 3
Your paper entitled “SIRT1, a master epigenetic regulator, and its role on DNA damage response in cancer” is a well-written review, complete and adequate in the cited reference work and complete in the scientific revision about the role of SIRT1 as a protective factor for cancer and as a tumor promoter. I suggest only few changes in English style and language:
- Rephrase the concept in lines 30-32 of Introduction (“Small DNA adducts…):
The sentence was corrected in the revised manuscript (lines 42-44).
- The same for the phrase in line 207 of the paragraph “Other forms by which SIRT1 modulates DNA repair response (“The absence of these factors…):
This sentence also was rewritten in the new manuscript (line 233-234).
- I also suggest a less twisted title:
The title was set to “The role of SIRT1 on DNA damage response and epigenetic alterations in cancer”
Round 2
Reviewer 1 Report
Authors have modified correctly the manuscript.
Author Response
Again, we are grateful for the suggestions, they were of extreme importance to this work.
Reviewer 2 Report
In the review SIRT1, a master epigenetic regulator, and its role on DNA damage response in cancer, the authors set out to summarize the role that Sirtuin-1 plays in DNA damage repair, and highlight the inconsistency of SIRT1 being alternatively upregulated or downregulated in different types of cancer.
This revision addresses a number of the specific concerns brought up previously, and overall the quality of the paper has improved. However, there are still a number of typographical errors within the manuscript that need to be fixed before it would be acceptable for publication. Again, a partial list, meant only to highlight the problem, is included below. The paper needs grammatical editing throughout. However, with proper editing, I feel this paper will be appropriate for publication.
Partial list of typographical errors:
Line 42: During theDNA synthesis
Line 57: of awide range of bulky DNA damage
Line 96: depends on the phase of thecell cycle
Line 141: we present evidences
Line 165: What is meant by “participate of an inactive heterochromatin state” ?
Author Response
Comments from the editor and reviewer:
Again, we are grateful for the important suggestions, which have been addressed in the new text. New revision in English was also conducted. Please, you will find below the detailed answers for your comments.
-Reviewer 2
Comments and Suggestions for Authors
In the review SIRT1, a master epigenetic regulator, and its role on DNA damage response in cancer, the authors set out to summarize the role that Sirtuin-1 plays in DNA damage repair, and highlight the inconsistency of SIRT1 being alternatively upregulated or downregulated in different types of cancer.
This revision addresses a number of the specific concerns brought up previously, and overall the quality of the paper has improved. However, there are still a number of typographical errors within the manuscript that need to be fixed before it would be acceptable for publication. Again, a partial list, meant only to highlight the problem, is included below. The paper needs grammatical editing throughout. However, with proper editing, I feel this paper will be appropriate for publication.
Partial list of typographical errors:
The typographical errors were corrected in the new text.
Line 42: During theDNA synthesis
Line 57: of awide range of bulky DNA damage
Line 96: depends on the phase of thecell cycle
Line 141: we present evidences
Line 165: What is meant by “participate of an inactive heterochromatin state” ?
It means that Sir proteins collaborate with the construction of an inactive heterochromatin state, but the sentence has been corrected in the text.